# Utilization of Fly Ashes from Fluidized Bed Combustion: A Review

**Katja Ohenoja** [1,]*, **Janne Pesonen** [2], **Juho Yliniemi** [1] **and Mirja Illikainen** [1]

[1]  Fibre and Particle Engineering Research Unit, Faculty of Technology, University of Oulu, P.O. Box 4300, 90014 Oulu, Finland; juho.yliniemi@oulu.fi (J.Y.); Mirja.illikainen@oulu.fi (M.I.)
[2]  Research Unit of Sustainable Chemistry, Faculty of Technology, University of Oulu, P.O. Box 4300, 90014 Oulu, Finland; janne.pesonen@oulu.fi
*  Correspondence: katja.ohenoja@oulu.fi; Tel.: +358-29-448-2395

**Abstract:** Traditionally fly ash is thought to be glassy, spherical particle originating from pulverized coal combustion (PCC) at temperature up to 1700 °C. However, nowadays fluidized bed combustion (FBC) technology is spreading quickly around the world as it is an efficient and environmentally friendly method. FBC is also able to utilize mixtures of low-grade solid fuels (e.g., coal, lignite, biomass, and waste) that have fluctuating quality, composition, and moisture contents. However, this leads to a high variation in the produced fly ash quality, unlike PCC fly ash, and hence challenges when attempting to utilize this fly ash. In this study, the utilization of fluidized bed combustion fly ash (FBCFA) was reviewed using the Scopus database. The most promising utilization target for FBCFA from biomass combustion is as a fertilizer and soil amendment. In construction, the FBCFA from various fuels is utilized as cement replacement material, in non-cement binders, as lightweight aggregates and cast-concrete products. Other types of construction applications include mine backfilling material, soil stabilizer, and road construction material. There are also other promising applications for FBCFA utilization, such as catalysts support material and utilization in waste stabilization.

**Keywords:** biomass ash; concrete; earth construction; fertilizer; recycling; woody ash

## 1. Introduction

The Traditionally fly ash is thought to be glassy, spherical particle originating from pulverized coal combustion (PCC) at temperature 1300–1700 °C. However, nowadays fluidized bed combustion (FBC) technology is becoming more popular as it is efficient and environmentally friendly. Unlike PCC technology, FBC technology can utilize mixture of low-grade fuels that have fluctuating quality, composition, and moisture content. Inside the FBC boiler, a sand bed is floating together with fuel on a forced high velocity air flow. The role of bed material is to improve the heat transfer and reduce temperature gradients ensuring a balanced combustion at a relatively low operating temperature of 700–900 °C. FBC also has less $SO_x$ and $NO_x$ emissions because of its lower burning temperatures and in situ capturing of $SO_2$ via direct reaction with Ca-based sorbents in bed material during the firing process:

$$CaCO_3 \Leftrightarrow CaO + CO_2 \tag{1}$$

$$CaO + SO_3 \Leftrightarrow CaSO_4 \tag{2}$$

According to [1], for every ton of coal burned in an FBC boiler, a 1/3 to 1/2 t of limestone is added to reduce sulfur emissions. This results in a three- to fourfold increase in solid-waste generation when compared with PCC. The ash originating from FBC is mainly—around 75–80%—fly ash of a fine-grain

size. Bottom ash, which corresponds to around 20–25% of FBC residue, does not generally constitute a disposal problem because it is extensively used as aggregate fill material for construction projects, filler in construction materials (wall board and dry wall), and de-icing solids for roads areas [2]. Therefore, in the current review, we concentrate only on fly ashes produced in FBC, abbreviated as FBCFA.

Around 14 million tons of FBCFA are generated annually only in US [3], and this amount is estimated to increase because of the construction of new FBC plants around the world. Moreover, in Europe and the United States, these plants are common; thus, FBCFA is produced in great quantities. FBCFA is utilized to some degree, but it still has unestablished utilization potential, with most of it being landfilled or disposed of. However, disposal is becoming more and more restricted and expensive. For example, in Finland, a tax price for one ton of FBCFA is 80 EUR. In addition, in populous countries such as India, there is a need to save agriculture and forest land from eventual fly ash dumping [4]. Hence, applications in which FBC fly ashes could be utilized efficiently have been studied widely and are reviewed in the current article.

The properties of FBCFA differ in many ways from PCC fly ash (PCCFA), which is widely adopted, for example, by the concrete industry [5–8], and have standardized properties [9]. PCCFA is a glassy, spherical shape pozzolan material that reacts with cement. FBCFA is more crystalline and irregularly shaped because of the hundreds of degrees lower burning temperature during the firing process (800–900 °C vs. 1300–1700 °C). FBCFA differs from PCCFA because of a high variety of fuel mixtures, additive possibilities, combustion temperatures, boiler technology (circulating, bubbling, pressurized, and atmospheric FBC) and fly ash collection technology. The most typical fuel for the FBC boiler is coal, but also coke, peat, biomass from forest and agriculture, and different types of wastes [10,11] are burned in a fluidized bed boiler.

Another issue to consider is the utilization of landfilled fly ash: this ash has reacted with water and, at some plants, mixed with bottom ash [12–15]. The utilization of landfilled ashes is one important issue to consider worldwide. However, in the present review, we have summarized fresh FBCFA properties, reviewed all possible utilization applications, and provided the most promising utilization applications.

## 2. Methods

The Scopus database was used to support the literature search because it is well known and the largest bibliometric information source for peer-reviewed studies. There are several keywords used to search for the appropriate literature. During the screening process, the words "fluidized/fluidised bed combustion" and "fly ash" were involved at all times. Together with those words, we checked the following specific utilization words: earth construction, soil stabilization, cement, concrete, mortar, construction, building, fertilizer, geopolymer, and alkali-activation, which were all checked using different writing styles. In addition to these well-known utilization destinations, we used more general search words to find all possible uses for FBC fly ashes, including utilization, reuse, exploitation, recycling, reclamation, salvaging, use, usage, valorization, and reutilization, which again were searched for using different writing styles. We chose only original articles written in English and published in journals to ensure high quality and appropriately peer-reviewed articles. After this, the abstracts of all references found from the screening process were read to see if those studies were related to FBCFA (not only spent bed material, for instance). The full-texts of the references found to be relevant were accessed using available online databases. The references selected for the current review were those with the above-mentioned terms (see Section 2) being included in either the article title, abstract, or keywords (including indexed keywords). The final selection of literature consisted references that are mostly from Europe and Asia. The number of FBCFA studies from different countries is presented in Figure 1.

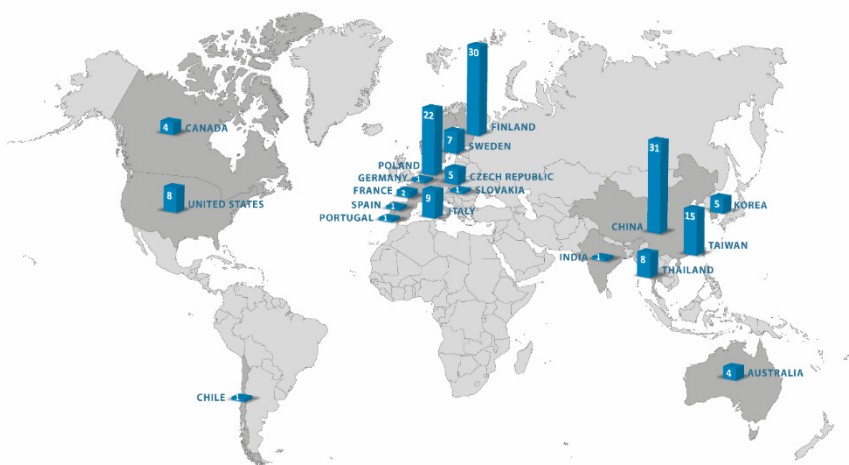

**Figure 1.** The number of studies on fluidized bed combustion fly ash (FBCFA) in different countries.

## 3. FBCFA Properties

### 3.1. Chemical and Mineralogical Properties

The chemical composition between the types of FBCFAs varies a lot; therefore, any conclusions made for one FBCFA may not be valid for another FBCFA. The main elements of FBCFAs are calcium (Ca), silicon (Si), aluminum (Al), sulfur (S), iron (Fe), and magnesium (Mg), and usually the contents of these elements are presented in oxide forms. The chemical composition of these components varies from almost 0 to over 50 wt % for CaO, $SiO_2$, and $Al_2O_3$, from 0.5 to 40 wt % for $SO_3$, and from 0.1 to 30 wt % for $Fe_2O_3$ (see Table 1).

**Table 1.** Chemical composition variety as the min and max values of the main components of the FBCFAs examined in this review.

|  | **Min. [%]** | **Max. [%]** |
|---|---|---|
| $SiO_2$ | 0.22 [16] | 53.5 [17] |
| $Al_2O_3$ | 0.10 [18] | 50.98 [19] |
| $Fe_2O_3$ | 0.10 [18] | 27.9 [20] |
| CaO | 1.40 [21] | 56.8 [22–29] |
| $SO_3$ | 0.50 [30] | 40.6 [31] |
| MgO | 0.15 [32] | 7.10 [33] |

The reasons why chemical composition of FBCFA varies, are combustion processes (e.g., how flue gases are treated) and different fuels. The typical chemical compositions of FBCFA from different fuel combustions are presented in Figure 2. The composition varies greatly depending on the fuel used. The difference in the $SiO_2$ content varies by 50%, in CaO content by 70%, in $Al_2O_3$ content by 40%, and in $Fe_2O_3$ content by 80% for fly ashes originating from waste and lignite combustion. In $SO_3$ content, there is an 83% difference between fly ashes originating from biomass (peat and wood) and coal combustion. In addition, fly ashes contain some minor elements, such as chloride (Cl), sodium (Na), titanium (Ti), phosphate (P), potassium (K), arsenic (As), cadmium (Cd), chromium (Cr), copper (Cu), mercury (Hg), nickel (Ni), lead (Pb), and zinc (Zn). As with the main elements, the content of the minor elements varies between all types of fly ashes. Phosphate and potassium are important trace elements for fertilization purpose, but the others are often detrimental to different utilization applications, for example, due to their possible environmental hazard and tendency to decrease concrete durability through deterioration of the microstructure.

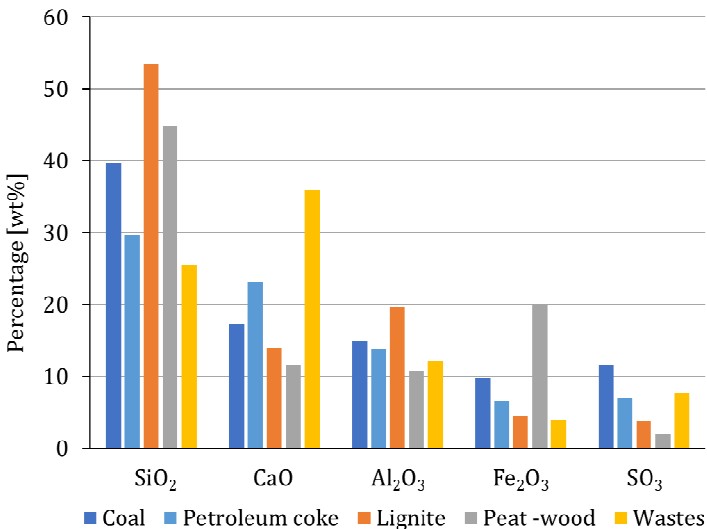

**Figure 2.** The Typical chemical composition of FBCFAs originating from different fuel combustion [30,34–37].

Moreover, the mineralogical composition of FBCFAs varies depending on fuel and the amount of chemical additions. The minerals found in FBCFA are quartz ($SiO_2$), anhydrite ($CaSO_4$), portlandite ($Ca(OH)_2$), lime ($CaO$), calcite ($CaCO_3$), and hematite ($Fe_2O_3$) (see Figure 3). Totally, 84 papers presented the mineralogical composition of fly ashes. As can be seen from the Figure 4, almost all fly ashes contain anhydrite (76 out of 84) and quartz (72 out of 84).

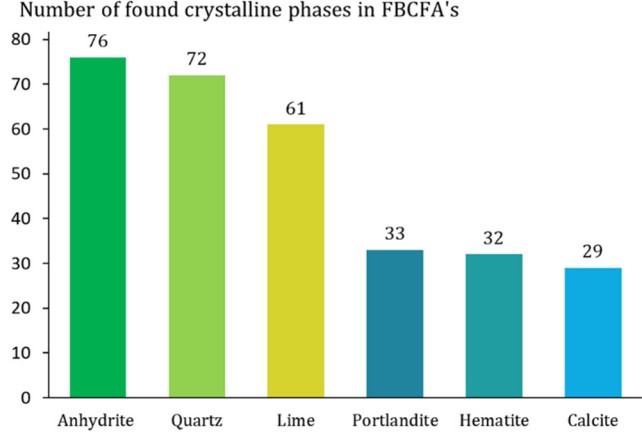

**Figure 3.** Crystalline phases found in FBCFAs (total number of FBCFAs examined was 84).

### 3.2. Morphological Properties

FBC fly ashes compose mainly of irregularly or angularly shaped particles [38,39] in contrast to tradiotional PCC fly ash which are mainly spherical [40] (see example from Figure 4). Spherical particles can be found also from some FBCFAs, but their share is much lower. This difference is due to the lower temperature of FBC technology (700–900 °C) compared to PCC technology at 1300–1700 °C: at FBC boiler temperature is not enough to melt irregularly shaped ash particles to spherical shapes like in the case of PCC. Moreover, fuel type [41] as well as type and amount of used adsorbent material can also effect on ash particle shape and its physical properties.

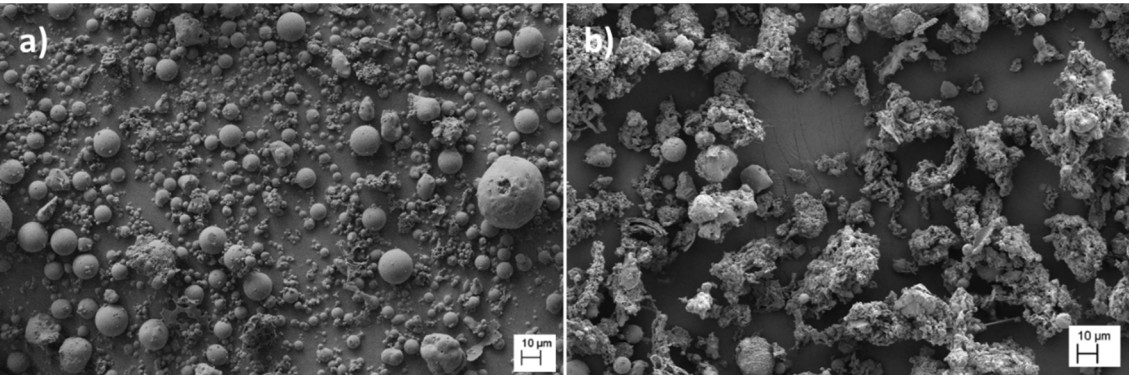

**Figure 4.** SEM images of (**a**) spherical particles in traditional PPC fly ash, and (**b**) irregularly shaped fluidized bed combustion fly ash.

### 3.3. Particle Size of FBC Fly Ashes

Particle size of FBCFA also varies, but it is usually slightly coarser compared with traditional PCC fly ash: around 20–30 μm (see Supplementary Materials). The smallest reported particle size is 1.63 μm for filter fly ash [42], and highest for paper sludge fly ash: 140 μm [30]. Avearge median particle size (d50) of FBCFA for arcticles reviewed here is 28.5 μm (totally 46 articles have reported the particle size). Fly ash collecting method has significant effect to particle size: fly ash origination from filter and last electrostatic precipitator is smaller than fly ash collected from silo or first elecrostatic precipitators. Based on the collected data from the literature (see Supplementary Materials), it seems that FBCFA from coal combustion has finer particle size compared to FBCFA from peat and wood combustion.

## 4. FBCFA as a Soil Amendment

### 4.1. Fertilizers and Soil Improvers

The In Nordic countries, there is a long history of utilizing fly ash as a fertilizer or soil improver in forestry and agriculture. This is mostly because biomass combustion is responsible for a considerable amount of total energy production in the forest-rich Nordic regions. Wood ash, in general, independently from the combustion method, contains all the nutrients which plants need in almost the correct proportions, excluding nitrogen, which is released into the atmosphere during combustion [43]. In Finland [44] and Denmark [45], there is national legislation regulating ash fertilizers (Table 2). The Finnish regulation sets the minimum acceptable content for Ca, the sum of phosphorous (P) and potassium (K), and the maximum content for detrimental elements; in the Danish regulations, only detrimental elements are regulated. Therefore, the excessive content of heavy metals [46] can prevent the use of fly ashes as fertilizers. In addition to national legislation, in the summer of 2019, the European Commission released a revision for the EU's fertilizer productions regulation [47]. The new regulation will be applied from 16 July 2022. New regulation opens the European market also for, e.g., organic fertilizers and recycled fertilizers.

**Table 2.** Finnish and Danish limit values for nutrients and harmful elements in ash fertilizers. These are the maximum allowed total concentrations of harmful elements (As, Cd, Cr, Cu, Ni, Pb, and Zn) and minimum concentrations of nutrients (Ca, K + P) using nitric-acid digestion.

| | As (mg/kg) | Cd (mg/kg) | Cr (mg/kg) | Cu (mg/kg) | Ni (mg/kg) | Pb (mg/kg) | Zn (mg/kg) | Ca (%) | K + P (%) | Neutralizing Value %Ca |
|---|---|---|---|---|---|---|---|---|---|---|
| Finland [a] | 25/40 | 2.5/25 | 300 | 600/700 | 100/150 | 100/150 | 1500/4500 | 6/- | 2 | -/10 |
| Denmark | | 5/20 [b] | 100 | | 60 | 120/250 [c] | | | | |

[a] Field fertilizers/Forest fertilizers. [b] Straw ash/Wood ash. [c] Straw ash or straw + wood ash/Wood ash used in forestry.

Most of the studies on ash fertilizers were conducted in Nordic countries, almost exclusively in Finland [48–62]. The typical viewpoint of most studies was to study the amount of pollutants in potential ash fertilizers and assess their environmental risks. Because ash fertilizers are in regular use in Finland, in the Finnish studies, the harmful elements and nutrient contents of the studied FBCFA were simply compared with the limit values of Finnish legislation. In most studies, a mixture of wood [49,55,63–65] or a mixture of peat and wood [48,50,52–54,59–62] were used as fuels in the FBC boiler; but in some cases, coal [66–70], sewage sludge [71], agricultural residue [72], swine manure sludge, olive kernel and olive pruning [73] and a co-combustion of sewage sludge and wood [74,75] were studied. Typically, the harmful element contents of the ashes were small and hence were fine for fertilizer use. There is also one study from Spain [63] where the amount of organic pollutants—polycyclic aromatic hydrocarbons (PAH), benzene, toluene, ethyl benzene, and xylene (BTEX) and styrene (S)—in wood bottom and fly ashes were studied. The conclusion was that the concentrations of the studied pollutants were low and did not possess environmental risks related to fertilizer use.

Even if the suitability of ashes for fertilizing purposes is typically evaluated using simple methods, more advanced analytical methods can be used to study the bioavailability of certain elements. Pesonen et al. [59] studied the effect of fly ash granulation on the bioavailability of harmful elements (As, Cd, Cr, Cu, Ni, Pb, and Zn) and nutrients (Ca, K, Mg, P, and S) in FBCFA fertilizers by using sequential extraction and aqua regia digestion. According to Finnish legislation, the FBCFA has to be processed (e.g., granulated) before application in fields or forests to prevent dust problems. The fractions of the sequential extraction were water-soluble fractions; exchangeable and acid-soluble fractions; reducible fractions; and oxidizable fractions. The first two were considered easily bioavailable, and the sum of all fractions were considered the total bioavailability. The results indicated that granulation significantly reduced the recoveries of the easily bioavailable nutrients Ca, K, Mg, P, and S, meaning that granulation reduces the quick-acting fertilizer effect of FBCFA. The results also indicated that the total bioavailability of Ca, K, Mg, and S was reduced after granulation. Therefore, the fertilizing effect of FBCFA is reduced after granulation. The total bioavailability of harmful elements was very low. The same observation regarding the leachability of heavy metals has also been made by Kuokkanen et al. [53] and Pöykiö et al. [62], who both used a sequential extraction procedure.

The fractionation of fly ash to improve its quality for fertilizing purposes has been studied in a few papers with the target to separate harmful substances from the fly ash [48–51,58,61,76]. FBCFA is typically collected using electrostatic precipitators (ESP); therefore, Ohenoja et al. [58], Dahl et al. [49], and Orava et al. [51] studied the use of electrostatic precipitators as classifiers of ash. Ohenoja et al. [58] found that the harmful elements (As, Cd, and Pb) were enriched in the finest ash fraction collected from second and third ESP fields (particle size $d_{50} < 18$ μm); therefore, the coarse ash from first field ($d_{50} > 18$ μm) would be better suited for fertilizer use. However, the authors did not study the behavior of nutrients such as K and P, but they noted that Ca was enriched to some extent in the fine ash fractions. Dahl et al. [49] studied the harmful element concentrations in paper mill fly ash compared with Finnish forest fertilizer limit values. Fly ash samples were collected from the first two ESP fields because all particulate matter was removed before the third field. Particle size distribution was determined using an automatic sieve shaker, and the smallest sieve was only <0.075 mm; therefore, 83.9 wt % of the particles from ESP 2 and 57.4 wt % from ESP 1 was in the smallest fraction. However, the harmful elements (As, Cd, Cr, Cu, Ni, Pb, Zn, and Hg) and nutrients (Ca, Mg, K, P, and S) were enriched in the field 2. According to Orava et al. [51], who studied FBCFA from four different power plants, harmful metals are enriched in the finest particles from field 3. The Cd concentration was, at best, fivefold more in the field 3 when compared with field 1. Pöykiö et al. [61] and Dahl et al. [50] studied the effect of sieving on the distribution of harmful elements in FBCFA. Harmful element (As, Cd, Cr, Cu, Ni, Pb, and Zn) concentrations were below the Finnish limit values; therefore, the FBCFA was suitable for use as a fertilizer. Moreover, in these studies, the harmful elements were mainly enriched to the smallest fraction. However, the FBCFA was only sieved to a particle size smaller than <0.074 mm, which

contained 70–90 wt % of the particles. Budhatjoki and Väisänen [48] found that particles smaller than 45 μm contained the highest concentrations of phosphorus and potassium, or 59 wt % and 45 wt % of the total amounts, respectively. However, the concentrations of harmful elements (As, Cd, Cr, Cu, Ni, Pb, and Zn) were also highest in the smallest fraction. Therefore, fractionation is not necessarily a good treatment for fly ash, even though the studied FBCFA's harmful element concentrations were below the Finnish limit values for forest fertilizers, even in the smallest fraction.

Mäkelä et al. [54] showed that it is possible to improve the quality of an FBCFA fertilizer by mixing it with another industrial side stream. The first mix use was paper mill FBCFA (15 wt %), desulphurization slag (15 wt %) from steel industry, sludge (25 wt %), and lime waste (45 wt %) from a paper mill. In a another study by Mäkelä et al. [55] that looked at forest soil amendment possibilities, the use of paper mill FBCFA (7.5–15 wt %) with blast furnace slag (7.5–15 wt %) from steel industry and sludge (30–45wt %), lime waste (30 wt %), and green liquor dregs (10 wt %) from a paper mill was researched. All manufactured fertilizers fulfilled the Finnish criteria for forest fertilizers, and some properties such as neutralizing capacity were comparable to commercial products [54,55]. Pesonen et al. [60] studied the co-granulation of FBCFA (40–100 wt %) with sewage sludge (0–40 wt %) and lime (0–30 wt %) for a fertilizer application. The nitrogen content increased as the sludge content of the granules increased, but it remained quite low (approximately 0.1–0.3 wt %) because of the relatively low dry matter content of the sludge. Moreover, some of the nitrogen was most likely vaporized as ammonia gas. The harmful element concentration was low, and the nutrient concentration was sufficient for forest fertilizer use according to Finnish legislation.

FBCFA has also been studied as a source of phosphorous for fertilizer purposes and has shown promising results. Ottosen et al. [71] and Pettersson et al. [74,75] studied the possibility of recovering phosphorus from FBCFA for fertilizer production. Ottosen et al. [71] used a two-compartment electrodialytic cell and obtained a 80–90 wt % recovery of phosphorus from sewage sludge FBCFA while the content of harmful elements (Zn, Cu, Pb, and Cd) was reduced to less than 20 wt % of the original amount. The obtained P-salt could be used as a raw material for fertilizer products. Pettersson et al. [74,75] studied phosphorus leaching from wood and sewage sludge co-combustion FBCFA for use in sulphuric acid extraction. The precipitation agent used in the water treatment plant affected the yield of phosphorus. When the Al-based agent was used 75–95 wt % of phosphorus was in the leachate, but when the Fe-based agent was used, 50 wt % was in the leachate. Moreover, the nutrients from the ash (Ca, K, and Mg) dissolved well into the leachate. However, Cd dissolved completely into the leachate, which can cause problems in fertilizer use.

### 4.2. Field Tests

FBCFA can be used as an effective soil amendment: soil pH and nutrient concentrations in a soil solution can be increased with the FBCFA addition to soil. Cruz et al. [64] studied the properties of the Portuguese bottom and fly ash and their application in soil uses even there are no regulatory guidelines for ash fertilizers in Portugal. A high total concentration of Cr was observed, but only a minor part was soluble on aqua regia. The application of ash caused an increase in the nutrients Ca, K, and Mg in the soil solution, but the solubility of Cu and Zn increased. FBCFA addition caused a short spike in the Cr concentration in pore water, which decreased over time. However, all concentrations of As, Ba Cd, Cr, Cu, Ni, Pb, and Zn were below the limit values for irrigation waters in Portugal. Ribeiro et al. [65] studied the effect that biomass FBCFA application has on soil fertility properties and plant growth. FBCFA addition (7.5 Mg/ha, dry basis) increased the soil pH to the recommended value, and the concentrations of Ca, Mg, and K increased in the soil. However, during the 60-day experiment, biomass growth did not increase after FBCFA application. Masto et al. [68] studied the co-application of coal FBCFA and sewage sludge on soil biological quality. The pH of the soil increased, which reduced the mobility of Zn, Co, and Cu, hence increasing the microbial activity. Clark et al. [77] conducted glasshouse experiments to test the effects of coal FBCFA additions to maize (*Zea mays* L.) growth in acidic soil (*Umbric Dystrochrepts*) and on maize shoot elements (Ca, S, P, K, Mg, Mn, Fe, Zn, Cu, and Al)

concentrations. They concluded that plants received the benefits when the FBCFA level was low (1–2%). Codling and Wright [78] studied the plant uptake of selenium, arsenic, and molybdenum from soil treated with coal FBCFA. They found that based on elemental concentrations in ryegrass shots and soil solution, selenium in FBCFA seemed to be a potential food-chain risk. Ribeiro et al. [79] tested FBCFA from a pulp and paper mill that uses bark and branches of eucalyptus as fuels. Other soil amendments were biological sludge from the pulp and paper mill wastewater treatment plant and commercial CaO. The studied soil was cambisol with a pH of 5.7 at the top layer and 5.1 at the bottom layer. FBCFA was either applied alone or 50:50 mixture (wt %) with sludge) using 7.5 Mg ha$^{-1}$ load. Soil pH was increased slightly (6.4 and 6.1 respectively) at the top layer but the pH in bottom layer was unaffected. CaO raised the pH to 6.7 at the top and to 5.5 at the bottom. Results showed that FBCFA nor sludge did not increase the mobility of heavy metals in the soil most likely due to the pH raise caused by ash addition. However, there were only small changes also in the availability of the macro nutrients, but both root size and plant height were increased when either FBCFA or FBCFA and sludge was applied. CaO addition decreased plant height compared to control. Moreover, Alvarenga et al. [80] studied eucalyptus (branches and bark) based FBCFA and biological sludge from a pulp and paper mill. The plant uses bubbling fluidized bed combustion technology (50 MW combustor). FBCFA was granulized with the sludge using mass ratios FBCFA:Sludge 9:1 and 7:3. Granules were used as fertilizers in different proportions to remediate soil that was affected by mining and contained high amounts of Cu, Zn, and Pb. Moreover, municipal solid waste compost was used together with the granules in some experiments. Granules improved the soil quality as the soil acidity was decreased leading to decreasing availability of Cu and Zn. Moreover, amounts of bioavailable P and K increased. However, in growth experiments using *Agrostis tenuis* considerable phytotoxic effects was observed most likely due to the low salt tolerance of the *A. tenuis* as the granule caused Cl$^-$ concentration to increase in the soil, therefore remediation was not achieved in these experiments. In addition to these, there are many Nordic studies concerning growth experiments using ash fertilizers. However, typically, these studies do not state the origin of the ash and were hence excluded from the current review. For example, Huotari et al. [81] made a review concerning the field trials of ash fertilizer studies.

FBCFA has been found to be effective as an acid soil amendment [82]. Moreover, the study of McCarty et al. [83] showed that FBCFA functioned as a soil liming material in a manner similar to that of CaCO$_3$. Riehl et al. [84] studied the addition of silico-aluminous (SiAl) and sulfo-calcic (SCa) FBCFA to a fluvisol (calcaric) to investigate the physical, chemical, and physico-chemical possible modifications of the amended soil; they found that SCa FBCFA seemed less well adapted to the integration in soils, mainly because of the basicity of the soil solution, which it induces. The SiAl FBCFA seemed better adapted from an agricultural and farming point of view because it can improve the cation exchange capacity, thus furthering the reserve of nutrients and adsorption of water as a useful reserve for plants. However, the authors found it lowered the structural stability of the soil. Rodak et al. [85] tested petroleum coke FBCFA that was combusted with 30 % CaO. The FBCFA was later mixed with limestone (33.3 wt %) and bottom ash (20 wt %) and used as soil improver in oxisol. FBCFA contained high amounts of Ni (0.77 g kg$^{-1}$) and the mixture was applied to achieve a dose of 0.9–1.6 kg of Ni per hectare. Growth experiments were performed on maize and soybean. Ni effects positively on the N metabolism in plants.but no improvement was achieved in the plant shoot weight even though the Ni concentration in shoot increased 4–5 folds and many parameters of the soil improved (pH, nutrient content).

There are also some studies on reducing phosphorus solubility utilizing FBCFAs. After nitrogen, phosphorus (P) has been one of the most limiting nutrients in most agricultural soils throughout the world. Loss of P from soils through runoff and leaching poses environmental degradation not only to land resources, but also to surface water (eutrophication) and ground water. Seshadri et al. [66,67] examined the effectiveness of FBCFA on phosphorus adsorption in soil. Their results indicated that P adsorption increased and leaching decreased with FBCFA additions. Moreover, the studies of Stout et al. [86–88] have shown that the solubility of P in high P soils can be reduced with coal FBCFA,

which decreases the potential for dissolved P export from the soils. There are also studies about adsorbing soluble P in animal manures using FBCFA. Animal manures contain large amounts of soluble P, which is prone to runoff losses when the manure is surface applied. The study of Dou et al. [89] showed that soluble P in animal manures can be adsorbed by mixing it with coal FBCFA. They found it desirable and beneficial because P availability for crops would be similar over the long term, whereas P runoff potential can be substantially reduced. Zhang et al. [69] studied the use of coal FBCFA (5–40 wt %) mixed with dairy, swine, or broiler litter manures to reduce P solubility to waterways. FBCFA addition reduced the solubility of P, mainly because of the high pH of coal FBCFA. The best result was obtained with the highest FBCFA addition (40 wt %). Atalay et al. [90] studied the possibility of immobilizing excess P in poultry litter with FBCFA. They found that FBCFA addition increased the yield of corn and soybean.

## 5. FBCFA as a Construction Material

### 5.1. Partial Replacement of Cement in Concrete

　　One—and the most studied—way to utilize FBCFA in construction is to use it as a partial cement replacement material to reduce the use of ordinary Portland cement (OPC). The utilization of FBCFA as a partial cement replacement material in paste, mortar, and concrete has been demonstrated in several studies, and the results are promising, even though there are some differing results most probably due to the high variation in FBCFA chemical composition (See Section 3). FBCFA has been found to have pozzolanic reactivity and hence the potential to partially replace OPC [38,91–98]. Typically, the initial setting time of mortars increases while the compressive strength decreases with an increasing amount of cement replacement by FBCFA; even Šiler et al. [42] found that the pastes containing FBCFA show higher strength, faster hydration, and a higher temperature in the first hours of hydration. Sinsiri et al. [99] found that the porosity and air permeability of FBCFA pastes are higher than those of PCC fly ash pastes. This was because of the higher irregular shape and surface of FBCFA compared with the spherical shape and the relatively smooth surface of PCC fly ash. Jozwiak–Niedzwiedzka [32] showed a significant influence of partial cement replacement by FBCFA on the chloride ion movements in concrete: They found that this kind of addition considerably reduced the chloride ion penetration. In addition to this, FBCFA has been found to have properties that possess the potential for antimicrobial applications [100].

　　The amount of OPC replacement by FBCFA was recommended to be limited below 20% by several authors [38,92,101–103] because FBCFA has little effect when its content is below 20%, but the strength decreases significantly if the ash content is over 20%. Nevertheless, Nocun-Wczelik et al. [104] noted that FBCFA acts as an excellent cement substitute when used as a 25% cement replacement, and Stryczek et al. [105] recommended a 30 wt % addition of mechanically activated FBCFA. Rissanen et al. [98] found that even with a 40% cement replacement ratio, the compressive strengths of the mortar samples were still as high as 88% of the control sample's strength.

　　The water requirement of cement has been found to increase when FBCFA is added [101]; however, it can be reduced by grinding [106,107]. The free lime (f-CaO) content of FBCFA has been found to increase the water requirement [101]. Song et al. [108] found that the polycarboxylate superplasticizer adsorption capacity of FBCFA-OPC paste is higher than that of PCCFA-OPC paste. Moreover, the adsorption amount of polycarboxylate superplasticizer increases with the ratio of ash to cement in the paste, and the fluidity of FBCFA-Portland cement paste is lower than that of the PCC fly ash paste. Their work indicates that when FBCFA is used as a concrete admixture, the poor flowability of the cementitious system because of the high adsorption of water and water-reducing agent should be taken into consideration.

　　Chen et al. [97] found that pastes containing FBCFA have a tendency to expand. They found ettringite (Aft) plays the main role in this expansion: Aft was generated mainly in 7 days, which has a large impact on the expansion development of the system at early ages, and the slight decomposition

of Aft in later days has little influence on the expansion. Chen et al. concluded that a grinding process should be preferred in the utilization of FBCFA: finer fly ash not only leads to a higher strength, but also contributes to a sharp development of expansion and reaches the maximum value as soon as possible. On the other hand, Wang and Song [109] suggested that the volume stability of a FBCFA cementitious system can be controlled effectively through autoclave curing or limiting the content of total $SO_3$ to below 3.5%.

Many FBCFAs contain gypsum, which is also used as a retarder in cement. The experimental results of Shen et al. [21] indicated that anhydrite in FBCFA can be used as an efficient setting retarder but can lead to lower mortar strength. Havlica et al. [94] found that there is no need to add extra gypsum when FBCFA is used as a cement replacement. In addition to partial replacement of OPC, the replacement of high-volume, low-calcium fly ash cement by high-sulphur FBCFA was also studied. Nguyen et al. [110] showed that FBCFA can significantly improve the mechanical properties at early ages because of more compacted microstructures. The accelerated hydration of tricalcium-silicate and more precipitated ettringite (Aft) in the formation of hardened pastes clearly confirm the important role of FBCFA in enhancing the mechanical properties of the modified high-volume, low-calcium fly ash (HVFA) cement pastes. Moreover, the addition of FBCFA did not influence the stability and passing and filling abilities of modified HVFA cement based SCCs [111].

Omran et al. [112] determined long-term in-situ performance of biomass FBCFA in concrete using full-scale concrete structural elements. They used 15–25% cement replacement level with FBCFA, and the results showed higher mechanical strength than the reference concrete made only with OPC. The concrete incorporated FBCFA decreased the permeability, and resulted in excellent durability.

*5.2. Non-Cement Binder*

5.2.1. Mixtures of Industrial Waste Materials

FBCFA is a suitable raw material for non-cement binders. FBCFA can be one of the raw materials for a ternary mixture of industrial waste materials consisting of ground-granulated blast furnace slag (GGBFS), low-calcium Class F fly ash, and FBCFA, which was called an SFC binder (S = GGBFS, F = type F fly ash, C = circulating FBCFA) by the authors. Chen et al. [24] studied the effect of different FBCFA amounts, and their results showed that 15–20 wt % FBCFA was the optimal value for a non-cement SFC binder to obtain the best compressive strengths. Nguyen et al. [23,25] studied the physical-chemical characteristics of paste and mortar with SFC cement, and they found that with a similar workability, the SFC mortars had compressive strengths and expansions comparable to that of OPC mortars [25] but had much better sulfate resistance than that of OPC mortars [23]. Nguyen et al. [29,113] also investigated the engineering properties and durability of high-strength, self-compacting concrete manufactured using an SFC binder, and they achieved 65 MPa after 28 days when 15–25 wt % FBCFA was used. Another non-cement binder in which FBCFA can be used is called SRF binder by the authors. This is also a ternary mixture of industrial side streams but contains GGBFS (S), FBCFA (F), and rice husk ash (R). Huynh et al. [31] evaluated the engineering properties and durability of a SRF mortar. They found that the mortar samples that incorporated 20% FBCFA and 30% rice husk ash exhibited the highest compressive strength values, and all of the samples exhibited good durability and high resistance to sulfate attacks. One more studied ternary binder was consisted of FBCFA, conventional fly ash and $Ca(OH)_2$ activator, and this paste yielded a compressive strength of 32 MPa after 28 days of standard sealed curing [114]. However, high free-CaO content of FBCFA may led to decreased compressive strength. Kang and Choi [115] determined that the optimal range of free-CaO content of FBCFA was 9–17%. Škvára et al. [116] found the use of polycarboxylate based plasticizer necessary for this ternary mix (FBCFA, conventional fly ash and $Ca(OH)_2$) in order to achieve similar properties to OPC.

In addition to ternary mixtures, FBCFA can be another raw material for a non-cement binder that contains only GGBFS (S) and circulating FBCFA (FA); this was called an SCA binder by the authors.

Dung et al. [22,26–28,117] studied this SCA binder in paste, mortar, and concrete. They found that SCA for paste and mortar has a proper setting and sufficient strength (up to 80 MPa at 28 days). The drying shrinkage of SCA paste was found to be lower than that of OPC paste [27]. The SCA mortars have been found to have good resistance to sulfate attacks, with a strength loss of about 15% or less [26]. The SCA concrete has a compressive strength at 91 days of approximately 50 MPa. In addition, SCA concrete shows moderate expansion at early ages and a low rate of shrinkage after 91 days of exposure [28]. Overall, SCA met the requirements for structural concrete [117]. Kledynski et al. [118] identified the hydration products present in hardened SCA pastes. Their results revealed the progression of the hydration process over time and the formation of products similar to those present in hardened cement pastes. The more FBCFA was incorporated in mixes, the higher the amount of bound water and the more porous the structure.

### 5.2.2. Self-Hardening of FBCFA

FBCFAs have a self-hardening property, meaning that when mixed with water, they produce the same type of reaction products as cement; therefore, they can be utilized as a single binder in low-strength concrete applications. Telesca et al. [119,120] investigated coal FBCFA as a raw material for the manufacturing of building components based on ettringite; they found FBCFA has the potential to be a material for this purpose. A maximum compressive strength value of about 6 MPa was reached at 70 °C and 16 h of curing. Nocun-Wczelin et al. [104] studied the hydration of FBCFA. They concluded that the hydration of FBCFA occurs continuously over 24 h, giving the heat higher values that are close to those for the mixtures with common cements because of the presence of soluble and other reactive compounds of the disordered structure. Li et al. [121] investigated the self-cementitious properties and pozzolanic reactivity of FBCFA. They also concluded that FBCFA is self-cementitious. At 28 days, they obtained a compressive strength of 11.4 MPa without chemical modifiers, and with chemical modifiers and particle size distribution optimization, they achieved 22.4 MPa.

In addition to FBCFA originating from coal combustion, also biomass fly ashes have a self-hardening property. The self-hardening property of FBCFA originating from the combustion of peat, wood, and waste was studied in three papers [20,30,106]. Illikainen et al. [20] found self-hardening strength to depend on the ashes' selectively soluble calcium, aluminum, and sulfate content, suggesting that the dissolution method is a fairly good means of assessing the hardening properties of FBCFA. Ohenoja et al. [30] found the compressive strength to depend on the selectively soluble calcium, aluminum, silicate, and sulfate content, even though the studied fly ashes differed greatly. The most critical elements were calcium and aluminum, and fly ash had an optimal ratio of Ca/Al that achieved over 10 MPa strength. Ohenoja et al. [106] studied the effect of grinding on the self-hardening of FBCFA originating from forest industry residuals and peat. The achieved compressive strength was 4 times higher (5 MPa vs. 20 MPa) after ball milling.

### 5.3. Aerated Concrete

FBCFA has been successfully used in aerated concrete production. Aerated concrete (known also as porous concrete), which can be used as a wall material, for instance, has many advantages, such as being lightweight, having a lower thermal conductivity, and being sound absorbent. It is usually made up of lime, cement, gypsum, and sand, with traces of aluminum powder as a pore-forming agent. The aluminum reacts with lime, which releases hydrogen gas and forms a large amount of tiny bubbles distributed uniformly over the matrix. Aerated concrete is typically cured in an autoclave (AAC), but also non-autoclaved concrete (NAAC) is produced. Łaskawiec et al. [122] studied the use of FBCFA in autoclaved aerated concrete and found that up to 40% of Class C fly ash from PCC can be substituted with FBCFA. Because of the phase composition of FBCFA, the content of lime can be reduced by 20% and the sulfate by 20–100 wt % in the concrete mix when compared with concrete mixes based on Class C fly ash. Chen et al. [123] utilized FBCFA for the preparation of foam concrete using steam curing at 60 °C for 24 h after demolding. They found the compressive strength, frost-resistance, and thermal

conductivity of the products to be good. They concluded that the strength of paste or foam concrete with stacked FBCFA is low, indicating that FA must be utilized as soon as possible. Song et al. [34] found that the maximum bulk density and compressive strength of AAC containing FBCFA are higher than those of AAC containing PCC fly ash, and the volume stability of AAC containing FBCFA is superior to that of AAC containing PCC fly ash. Song et al. [124] confirmed that autoclave curing can effectively limit the expansion of anhydrite in FBCFA. Wang et al. [19] used metallic aluminum in MSWI bottom ash to make AAC that uses FBCFA. The extent of air entrainment in the resulting AAC specimens varied by changing the bottom ash to fly ash ratio. Their study proved that AAC with satisfactory properties can be successfully prepared from a combination of MSWI bottom ash and FBCFA. Glinicki et al. [125] found that the utilization of FBCFA resulted in a degradation of the frost salt scaling resistance of aerated concrete. They found the effect to be pronounced for increasing cement replacement levels and for increasing the unburned carbon content of FBCFA.

As in AAC, FBCFA has been successfully utilized in NAAC and in air entrainment concrete. Xia et al. [126] utilized FBCFA in non-autoclaved aerated concrete and their study showed that FBCFA can be used as raw materials in NAAC production at a proportion of around 65%. Lee and Lee [127] studied lightweight foamed concrete prepared from FBCFA, which a high free CaO amount was stabilized using carbonation. They replaced cement 30 wt % and stored more than 90% of the $CO_2$. Glinicki and Zielinski [128,129] studied the influence of FBCFA on the microstructure of air voids in hardened air entrainment concrete. They concluded that proper air entrainment of concrete made with the addition of FBCFA was possible to produce although it required a significantly increased amount of air entraining admixture [128]. They also found that the cyclic freeze–thaw exposure did not have any significant influence on the phase composition of concrete with and without the FBCFA. Jozwiak–Niedzwiedzka [32] showed that FBCFA in air entraining concrete reduced the chloride diffusion coefficient.

### 5.4. Alkali-Activated FBCFA

Alkali-activation of FBCFAs has been studied much less than PCC fly ashes, for which the number of studies is extensive. Most investigations related to alkali-activation of FBCFAs have been conducted in Thailand, South Korea or Finland. It can be stated that FBCFA could be suitable for an alkali-activated material precursor, but often co-binders or pre-treatment of fly ash is needed. The studies acknowledged the low reactivity of FBCFAs in alkali-activation due to their crystalline nature, but compressive strengths of 10–30 MPa were often achieved [130–139], which is enough for most mortar- and paste-type applications. Even up to 50 MPa was reached [140] when most of the binder was slag-based and fly ash amount was 10–20 wt %. The most common co-binder was metakaolin [33,37,138,141,142], but also GGBFS, PCC, OPC and FBC bottom ashes were used [37,130,132,135,136,141–144]. Pre-treatment by grinding [132,133,136,137,139,145,146] or alkali fusion [143,146] was used in some papers to increase the reactivity of the ashes. In almost all the investigations, Na-silicate and/or NaOH was the alkali-activator of choice, but also sodium carbonate has been studied [147]. Curing was conducted typically at 20–60 °C for 1 or 2 days. The phases that formed in alkali-activation were mostly amorphous including calcium-(aluminate)-silicate phases due to the high Ca content in FBCFAs.

FBCFA can also be used as an alkali-activator. Hsu [148] studied the influence of FBCFA and different alkalis on the properties of alkali-activated slag cement mortars. Hsu concluded that when FBCFA was used as an alkali-activator, the workability of the mortar specimen declines, the bleed water reduces, water adsorption increases, and drying shrinkage decreases as the amount of added ash increases. Hsu also found that a 10% addition of FBCFA increased early strength, but 20% and 30% additions decreased when compared with the control. Wu et al. [149] presented the feasibility of FBCFA as an alkali-activator for GGBFS; they concluded that the optimum mixing ratio of FBCFA and GGBFS was 30%:70%, respectively, reaching up to 31 MPa compressive strength.

## 5.5. Lightweight Aggregate Production

FBCFA can be used as a raw material for lightweight aggregates for concrete. Kratochvíl et al. [150] prepared alternative aggregates based on FBCFA which resulted in concrete with compressive strength over 25 MPa, and there is no natural origin aggregate having such values. Shon et al. [151] used FBCFA as a primary resource material for manufacturing synthetic lightweight aggregate, and the results indicated that the produced synthetic aggregates met ASTM C 33/330 criteria for a concrete aggregate without inducing any soundness problems, despite their relatively high water absorption. In addition to coal FBCFA, Yliniemi et al. produced strong lightweight aggregates by granulating peat-wood FBCFA [37], recovered fuel-biofuel FBCFA [142], and FBCFA and mine tailings [152]. The aggregates produced met the definition of lightweight aggregate in standard SFS-EN 13055-1, and the results showed that alkali-activated aggregates had physical properties comparable to commercial lightweight expanded clay aggregates (LECAs). Mortar and concrete prepared with alternative aggregates had higher mechanical strength, a higher dynamic modulus of elasticity, and higher density than concrete produced with LECAs, while the rheology and workability were the same [152].

## 5.6. Cast-Concrete Products

FBCFA can be used in cast-concrete products. Naik et al. [153] produced cast-concrete product specimens containing a maximum 40 wt % of FBCFA, and these specimens exceeded the minimum compressive strength requirements of ASTM from early ages. The abrasion resistance of paving stones made by replacing up to 34% of cement with FBCFA was equivalent to that of the control paving stones. The cast-concrete products made by replacing up to 40% of cement with FBCFA were equivalent in compressive strength (89–113% of control) to the products without ash. FBCFA can also be used to make high-quality bricks and wall and floor tiles. Zhang et al. [154] showed that an autoclaved brick could be made up of 77% FBCFA, 20% FBC bottom ash, and 3% cement by weight, and it exhibited good long-term volume stability and achieved a compressive strength of up to 14.3 MPa. Chen et al. [155] manufactured the unfired bricks using FBCFA by dry pressure molding method. Their results showed that the compressive strength of the brick reached the highest level (23 MPa) when it was manufactured by cement 8%, bottom ash 15%, lime 12%, gypsum 5%, FBCFA 60%, and water 15%. Húlan et al. [156] investigated the influence of the FBCFA content in illite-based ceramics on the ceramic's thermophysical and elastic properties during the heating and cooling stages of firing. They found 30 wt % of FBCFA in the mixture to be the upper limit if the maximum firing temperature was 1100 °C without a soaking time being used. They found a positive effect of FBCFA when added to ceramics, which showed to have lower shrinkage during the firing process. On the other hand, FBCFA caused a deterioration of the ceramic's mechanical properties, which is reflected in the values of Young's modulus and its mechanical strength.

## 6. FBCFA in Earth Construction

### 6.1. Mine Backfilling

Subsidence and acid mine drainage are two major problems found in coal mining areas [157]. There are few studies about the use of FBCFA in mine backfilling [17,157–161]. In this chapter, we review only mechanical filling subjects, and mine waste water treatment is reviewed in Section 7.6. Chugh et al. [158] developed a pneumatic backfilling technique that was be capable to fill underground voids. The used FBCFA as the backfilling material for abandoned mines. Siriwardane [157] showed that a grout made of FBCFA can be successfully placed in a mine cavity by hydraulic backfilling to reduce the problems caused by subsidence. Solem–Tishmack et al. [159] found FBCFA to be applicable for high-volume utilization in engineered structures and capable of immobilizing boron and selenium with solidification/stabilization (S/S) technology. Pomykala et al. [35] successfully FBCFA-water suspensions in underground coal mines to fill the voids and cavings resulting from exploitation. Kepys et al. [160]

found that almost all of the suspensions prepared from FBCFA from agro-forestry biomass, cement, and water can be applied in mines for gob caulking.

*6.2. Soil Stabilization*

One interesting way to utilize FBCFA is as a soil stabilizer. There are two studies about this use type, but there are definitely more cases not scientifically published. Shao et al. [162] found in a strength test that the compressive strength of FBCFA stabilizing soil is 3–5 times that of pulverized fuel fly ash (PFA) for stabilizing soil at 28 days. They found that when the stabilizer's total content was 10% and the amount of FBCFA was 72%, the compressive strength of the stabilized soil can reach the maximum of 2 MPa at 28 days of curing. The stabilizer can meet the strength requirements of a cement-soil mixing pile composite foundation and of a cement-soil mixing pile waterproof curtain. Lake sludge is one typical byproduct material of lake or river dredging in China [163]. Tang et al. [163] treated the dredged lake sludge by using solidification technology and mixing FBCFA with cement or lime; they found that at the same curing time and content of the stabilization materials, the strength of the specimens decreased as the FBCFA/cement ratio increased, and the optimum ratio between FBCFA and cement was found to be 2:3.

*6.3. Road Construction*

FBCFA can be successfully used as a road base material in the form of binder or aggregate [1,164–167]. Shon et al. [164] used aggregates produced from FBCFA for road base course construction. Based on their strength, swelling, and tube suction test results, a mixed design containing 10% fresh FBCFA and a 50/50 blend of fresh and stockpiled FBC ash satisfied the requirements of a Class M base, as per TxDOT Item 276. Mráz et al. [167] studied the use of FBCFA as an alternative binder in subgrade structures and the roadbed materials of roads, as well as in structural pavement layers. They concluded that mixes including FBCFA met the minimum value required for simple compressive strength according to technical specifications in the Czech Republic; here, the volumetric changes were small. However, they concluded that these mixes were not resistant against frost and water. Moreover, White and Bergeson [168] found signs of freeze-thaw durability problems when FBCFA was used as a road base material. Jackson et al. [165] presented an application to recycle both the bottom ash and fly ash from FBC boilers as pavement base course materials. Their laboratory testing and field application results suggested that the hydrated ash has a stiffness equal to or greater than that of lime rock. Deschamps [1] found swelling to be the primary limitation when using FBCFAs. Therefore, he recommended to stockpile the fly ash with exposure to moisture for a long period prior to use. Yoon et al. [169] continued studying the swelling; during and after the embankment construction, heaving of the embankment with 60% FBCFA, 35% stockpiled ash, and 5% Class F fly ash was observed. They concluded that the use of FBCFA as a fill material requires caution because of potential excessive swelling from ettringite formation. Moreover, Yoon et al. recommended FBCFA to be exposed to moisture before utilization. The study of Iwanski et al. [166] revealed the potential of FBCFA to replace the cement used as a binder in the bound base layers. The optimum quantities of the additions for the base layers with dolomite was 25% of FBCFA.

FBCFA has been found to be more suitable for asphalt concrete than virgin mineral powders. Li et al. [170] investigated the potential of utilizing FBCFA as an alternative filler, substituting mineral powders that are widely used in asphalt concrete. Their results showed that generally FBCFAs have a greater effect on improving the performances of asphalt and that the specific surface area, free CaO, morphology, and mineralogical phases of the FBCFAs are more favorable than those of the mineral powders, respectively, while the alkaline values, hydrophilic coefficients, particle size distributions, and water contents of the two fillers are similar.

## 7. Other Applications of FBCFA

### 7.1. Recovery of Combustibles

Some FBCFAs contain a high amount of unburnt carbon, which causes energy loss, prevents their utilization in concrete, for instance, and increases the fly ash total amount. Jain et al. [4] studied two different FBCFA samples to explore the scope of the recovery of combustibles. The first ash sample did not show any recovery potential, but the second ash sample indicated that about 40% of the material could be recovered, with 35% fixed carbon and a 10,841 kJ $kg^{-1}$ gross calorific value. They found that recovered ash can be used as a fuel blend in standard fluidized bed combustion boilers for efficiently burning inferior coal.

### 7.2. $SO_2$ Capture

One possibility for FBCFA recycling lies in a hydration process aimed at reactivating the $SO_2$ sorption ability of the unconverted lime. Typically, calcium-based sorbents (calcitic limestone and dolomite) are used for capturing $SO_2$ in FBC systems that burn high-sulfur fuels. One of the major limitations of this technology is the relatively low utilization of sorbent (30−40%); therefore, excess limestone sorbent is required to achieve an acceptable $SO_2$ capture efficiency. Montagnaro et al. [171] found the reactivation of sorbent material by water hydration of FBCFA to be effective for the production of a good-quality sorbent for in situ sulfur capture during FBC. They found that hydration at 70 °C for 48 h of the raw fly ash yields an ettringite-rich material, and ettringite represents the main source of free lime available for sulfur uptake as the reactivated ash is calcined. Bernardo et al. [172] studied ettringite in more detail because it can play a chemical and physical role in capturing $SO_2$. They investigated the conditions under which ettringite is formed by the liquid-phase hydration of FBC waste. Wu et al. [173] hydrated five FBCFA with liquid water or steam to determine whether hydration could improve sorbent utilization in these samples under FBC conditions. After hydration, for two fly ashes and three carbon-free samples, the capacity for taking up $SO_2$ showed a limited or medium improvement; however, hydration was evidently ineffective in reactivating the remaining samples. Therefore, the authors suggested other techniques to be considered, such as mixing and pelletization with bed material. Later, Wu et al. [174] presented a new technique for the reactivation of FBC-spent sorbent and preparation of pellets suitable for capturing $SO_2$, which can also incorporate the fly ash into the pellets so that it has an adequate residence time in the primary combustion loop of a CFB to realize improved sulfur capture; they found reactivated pelletized sorbents to show an improved sulfation rate when compared with both the original sorbent and the spent sorbent, particularly during the diffusion-controlled reaction stage. Baek et al. [175] fabricated Ca-based in-furnace desulfurization sorbets utilizing FBCFA, and they found excellent desulfurization performance.

### 7.3. $CO_2$ Sequestration

FBCFA is also found to be good for capturing $CO_2$ at FBC plants. The carbonation rate of the FBCFA is found to be the same as that of $Ca(OH)_2$ [176]. Patel et al. [16] examined if FBCFA with a limestone feed can capture a part of the $CO_2$ released from it; their results showed that the carbonation reaction in lime followed a similar pattern as in fly ash. They concluded that a partial $CO_2$ sequestration by untreated hydrated fly ash could be an effective option for capturing and sequestration $CO_2$ from FBC plants with sorbent injection. Moreover, the results of Jaschik et al. [177] clearly showed that FBCFA originating from lignite combustion has the highest potential for carbonation (compared with PCCFA) because of its high content of free CaO and fast kinetics of dissolution and that it can be employed in mineral carbonation of $CO_2$. Moreover, Bae and Lee [176] found FBCFA to be good for capturing $CO_2$.

### 7.4. Adsorbents and Catalysts

FBCFA has been found to be a good raw material for adsorbent materials or catalyst support. Ruiz et al. [178] used FBCFA originating from forest biomass combustion as a precursor of porous silica materials for environmental applications. They selected a sieved fraction with a grain size >500 μm as a precursor material for chemical activation with KOH and high temperature (750 °C) because that fraction had the highest content of unburned carbon and particularly good texture properties. They found the adsorbent-catalyst materials to be mainly mesoporous materials with a significant contribution of micropores. FBC coal fly ash was found to be a good raw material for the synthesis of zeolites, which are crystalline aluminosilicate compounds that can act as sorbents, catalysts, and ion-exchange materials [179]. Grela et al. studied two methods of zeolite synthesis using FBCFA as the raw material. These syntheses were fusion synthesis and low-temperature synthesis. The materials obtained by means of low-temperature synthesis have a BET-specific surface area amounting to about 213 $m^2$/g, and those obtained by means of fusion synthesis have a BET surface area amounting to about 120 $m^2$/g.

### 7.5. Filler Material in Polymer Composites

There is a potential for recycling FBCFA as a suitable filler material in polymer composite materials. Yao et al. [180] studied the use of FBCFA to improve the toughness of the composite. FBCFA was coated by stearic acid and used in a composite of polypropylene/ethylene vinyl acetate/high density polyethylene (PP/EVA/HDPE) using a molding process method. They found that the tensile strength and elongation at break of a stearic acid coated with an FBCFA composite were greater than that of an FBCFA composite alone, and the elongation at break of a stearic acid coated FBCFA composite increased 30% compared with the composite without fly ash. Garbacz and Sokołowska [181] showed that standard quartzite microfiller in polymer-cement composites is possible to replace with FBCFA of hard coal combustion but in a limited range due to the decrease of mix workability which resulted in deterioration of composite properties. When FBCFA replace was up to 50% of microfiller (10–12% of total composite mass), the values of technical properties are still high (decrease by ~15%).

### 7.6. Acidic Wastewater Treatment

Acid mine drainage is one major problem around mining areas. These acid wastewaters may cause severe impacts on the environment acidifying and discharging large amounts of salts and heavy metals into aquifers. Grouting an abandoned mine with alkaline materials provides a permanent reduction in acid production. Among the various acid mine drainage remediation options, the most used and cost-effective is the addition of neutralizing agents such as magnesite, limestone, and dolomite. The results of Siriwardane et al. [157] showed that a grout made of FBCFA can be successfully placed in an underground coal mine cavity using hydraulic backfilling and to reduce the acid mine drainage. González et al. [18] studied the potential use of FCBFA in neutralization and heavy metals removal from acid wastewaters to reduce the use of natural resources as raw materials; their study showed a high removal efficiency for $Cu^{2+}$, $Pb^{2+}$, and $Cr(VI)$ and to neutralize acid wastewaters. Later, González et al. [182] developed a non-conventional sorbent using a blend of FBCFA and the organic waste fraction from a Kraft cellulose wastewater facility. Their neutralization and heavy metal fixed-bed removal tests showed that the produced blend is a suitable sorbent material at pH 4, showing great buffering and $Cu^{2+}$ and $Pb^{2+}$ removal capabilities.

### 7.7. Waste Stabilization

Hazardous waste with a high amount of heavy metals are often made non-hazardous by solidification/stabilization (S/S) treatment, which is traditionally done with cement and lime. The main purpose of waste disposal stabilization is to immobilize pollutants into a matrix, allowing for a safer way for landfill disposal of that waste. FBCFA as a one stabilizer binder has been found for many

different materials: municipal solid waste (MSW) fly ash [183,184], metal-bearing sludges [185], hot dip-galvanizing ash with a high content of zinc [186], galvanic sewage sludge [187], and coal slimes [188]. On the other hand, FBCFA originating from waste combustion tends to have a high amount of leachable heavy metals, which are needed to stabilize to cement or geopolymer matrices. Pesonen et al. [189] studied the ability of S/S of recovered fuel-biofuel FBCFA, which showed the high leaching potential of Cr, Mo, Pb, and Zn. The binders used were Portland cement, an alkali-activator (sodium silicate + sodium hydroxide), and a mixture of the two. The S/S efficiency of Pb and Zn was >96.3% and >94.5%, respectively. However, the authors found increased leaching of Cr and Mo, especially after alkali-activation. Therefore, they recommended simultaneous use of OPC and alkali-activation for the S/S process. Barbosa et al. [190] studied the S/S of FBCFA originating from coal, sewage sludge, and meat and bone meal. All S/S mortars achieved a good degree of stabilization characterized by medium to high compressive strengths and being classified as non-hazardous materials, according to their leaching behavior [190].

## 8. Conclusions and Outlook

There is a wide variety of possible utilization applications for FBCFA (Table 3). One of the most promising applications is fertilizers for fly ashes originating from biomass combustion. For those ashes that cannot be used as a fertilizer, utilization as an alternative binder material is the best route because of the self-cementitious property of FBCFAs. There is a lot of encouraging research when it comes to utilizing FBCFA in non-structural applications, such as aerated concrete or lightweight aggregates. The utilization level of fly ashes can be further improved by mechanical treatments, such as grinding and fractionation. Typically, FBCFA does not meet cement and concrete standards, so it is not possible to utilize FBCFA in standardized cement or concrete. For example, European fly ash utilization standard EN 450-1 applies to ashes originating from pulverized combustion and where the coal content must be over 60% or over 50% when coal combustion takes place with pure wood. However, it is reasonable to expect that in the future extension of the current regulations on using FBCFA in concrete should be limited to those fly ashes meeting the physical and chemical requirements specified in the standards. The chemical composition between FBCFAs varies a great deal. The main components of FBCFAs are $SiO_2$, $Al_2O_3$, CaO, $Fe_2O_3$, and $SO_3$, and the composition between those components varies from 0.1 up to over 50 wt %. Therefore, quality evenness is the foremost challenge for FBCFA complete utilization. However, this can be solved by opening ash treatment plants in which quality evenness can be ensured.

**Table 3.** Variety of the possible utilization applications of FBCFA.

| Application | Number of Studies | Limitations/Important Information etc. |
|---|---|---|
| Soil amendment | 46 | - Only for fly ashes from pure biomass combustion<br>- Low amount of hazardous elements<br>- Source for phosphorous and calcium |
| Construction | 67 | - Chemical composition often suitable for cement substitution<br>- Usually fly ashes have pozzolanic activity<br>- Contain reactive minerals (portlandite, lime, anhydrite)<br>- Irregular shape → higher porosity<br>- Initial setting time usually increases |
| Earth construction | 14 | - Mine backfilling material<br>- Immobilizing some hazardous elements via stabilization/solidification method<br>- Decreased resistant against frost and water |
| Combustion plant internal use | 8 | - Re-combustion of fly ashes containing high unburnt carbon<br>- $SO_2$ and $CO_2$ capture with lime-containing fly ashes |
| Other | 13 | - adsorbents, filler material, waste stabilization, ... |

**Supplementary Materials:** The data collected from the literature is available in excel sheet with references. It is available online at http://www.mdpi.com/2071-1050/12/7/2988/s1.

**Author Contributions:** Conceptualization, methodology, formal analysis, investigation, K.O., J.Y., J.P. and M.I.; writing—original draft preparation, K.O., J.Y., J.P.; writing—review and editing, J.Y., J.P., M.I.; visualization, K.O.; funding acquisition, M.I. All authors have read and agreed to the published version of the manuscript.

**Funding:** This work was done under the auspices of the ARCTIC-ecocrete project, which is supported by Interreg Nord EU-program and the Regional Council of Lapland.

**Conflicts of Interest:** The authors declare no conflict of interest.

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
