# Peer review of "Utilization of Fly Ashes from Fluidized Bed Combustion: A Review"

_sustainability, doi:10.3390/su12072988_

Round 1

Reviewer 1 Report

The paper contains the review on the subject of use of fluidized fly ashes, i.e. the fly ashes from FBC of various fuels. The authors revised the Scopus database.

The work is extensive and quotes a lot of papers that have been published in English. The authors say that they searched for papers by keywords - mainly "fluidized/fluidised bed combustion" and "fly ash". However, at the beginning of the introduction of fluidized bed technology there was no fixed terminology regarding fly ashes from FBC installations. A lot of scientific work concerns such fly ashes, but scientists have tried to describe them with the previously known nomanclature and often referred to their chemical composition (especially the increased amount of calcium compounds  due to the use of calcium sorbents), therefore they used terms such as "calcareous fly ash" or "calcium fly ash". The work may have exclude these papers even though they are present in the Scopus database. Nevertheless, the current list of references allows to state that the research on the state of knowledge was carried out very well.
The manuscript is generally well written.
After a number of additions and minor corrections the article can be published.

Introduction section: For readers unfamiliar with FBC technology it could be explained in more detail how it differs from conventional (pulverized) coal combustion - not only in terms of the temperature, but also the construction of furnaces/instalations. Especially since the combustion scheme appears in the graphic abstract (line 30).

Regarding the content of the paper, some data could be updated. For example in line 53 the authors provide the amount of FBCFA produced annually in the PRC, while it is data from 2014. It can be expected that much more fluidized fly ash was produced in 2019.

In section 3 the attention is paid to the chemical and phase composition of the FBCFA, but information on the morphology of particles of fly ashes from the FBC of various fuels is missing. The authors mention that FBCFA particles are not as perfectly spherical as siliceous fly ash particles in the beginning of the paper (and in the abstract), but they do not discuss the morphology. There is also a lack of information about the of particle size of fly ash, about the high content of particles smaller than 1 micron, resulting in developed specific surface area. It is worth supplementing Section 3 with micrographs of various fluidized fly ashes or examples of size distribution plots.

The term “non-cement binder” used in the title and text of section 5.2 (lines 375, 377, etc.) and in the graphical abstract is a bit vague. Wouldn't it be better to write "cement-less binder"?

The title of section 7.3 “CO2 capture” could be changed to “CO2 sequestration”. Also in the same section the term “carbonization” should be changed to “carbonation” which is more common term describing the neutralization of calcium hydroxide by carbon dioxide in the presence of moisture/water in the concrete technology.

Placing the table in the Conclusions section seems to be a bit unusual. Perhaps the authors could present those data in a more traditional form of the text.

Reviewer 2 Report

The paper is a very interesting example of a review paper, with precisely described references. The only comment to the authors is to make better diversification of the original sources (regarding European countries) and add some more sources from the US (the number is relatively small).

Author Response

The authors wish to thank the Reviewer for his/her comments on our work. We truly appreciate your time and the opportunity to correct the manuscript.

We have made our data collection in Scopus database and used the search words listed in Methods section. Based on the search results, unfortunately we found only few articles in which FBC fly ashes from US was studied. Reason for this may be different terminology: in this paper we concentrated only to papers highlighting the combustion technology being FBC, not fly ashes with some special chemical composition or fuel for instance.

Reviewer 3 Report

A fine review recognizing potential issues of many kinds, including health

Author Response

The authors wish to thank the Reviewer for his/her comments on our work. We truly appreciate your time and opportunity to correct the manuscript.